# Clinical Value of CT for Differentiation between Ascites and Hemorrhage: An Experimental In-Vitro Study

**DOI:** 10.3390/jcm10010076

**Published:** 2020-12-28

**Authors:** Maximilian Kerschbaum, Leonhard Andreas Schurr, Moritz Riedl, Agnes Mayr, Isabella Weiß, Lisa Klute, Daniel Popp, Christian Pfeifer, Antonio Ernstberger, Volker Alt, Lena Marie Dendl

**Affiliations:** 1Department of Trauma Surgery, University Medical Center Regensburg, Franz-Josef-Strauss-Allee 11, 93053 Regensburg, Germany; moritzriedl@icloud.com (M.R.); agnes-mayr@t-online.de (A.M.); isabella.weiss@stud.uni-regensburg.de (I.W.); Lisa.Klute@ukr.de (L.K.); daniel.popp@ukr.de (D.P.); christian.pfeifer@ukr.de (C.P.); volker.alt@ukr.de (V.A.); 2Department of Surgery, University Medical Center Regensburg, Franz-Josef-Strauss-Allee 11, 93053 Regensburg, Germany; Leonhard.Schurr@klinik.uni-regensburg.de; 3Department of Trauma Surgery, Clinic Osnabrück, Am Finkenhügel 1, 49076 Osnabrück, Germany; antonio.ernstberger@klinikum-os.de; 4Department of Radiology, Johanniter Hospital Treuenbrietzen, Johanniterstraße 1, 14929 Treuenbrietzen, Germany; lena_marie_dendl@yahoo.de

**Keywords:** ascites, abdominal injuries, trauma, hemorrhage, computed tomography

## Abstract

Background: Abdominal trauma, leading to intra-abdominal bleeding, is a life-threatening condition that might need emergency surgical intervention. Sonography and Computed Tomography (CT) are most commonly used to detect free intra-abdominal fluid. This study investigates the accuracy of CT to distinguish between ascites and intra-abdominal hemorrhage. Methods: Ascites were collected during a clinical routine. Three serial dilutions, mixing ascites with whole blood samples of the patient and with two blood group identical donors, were prepared. Laboratory-chemical analysis and radiological evaluation using CT with measurement of average Hounsfield Units (HU) were performed. Results: Between ascites and whole blood as well as between ascites and the 1:1-ratio-samples, HU values differed significantly (*p* < 0.001). All further dilutions showed HU values with no significant difference compared to ascites (*p* ≥ 0.42). Whole blood showed significantly higher HU values than ascites and every step of the serial dilutions (*p* < 0.001). Measured HU values were also dependent on time and the exact point of measurement in the micro reaction vessels. Conclusions: In patients suffering from blunt abdominal trauma with preexisting ascites, HU values in CT imaging are not valid enough to exclude an acute hemorrhage.

## 1. Introduction

Abdominal trauma is a serious condition that is present among 7–10% of all trauma patients and often requires surgical intervention [1,2]. Abdominal trauma victims account for up to 24.1% of overall mortality rates. High mortality rates occur, especially among severely traumatized patients with an Injury Severity Score (ISS) of more than 15 [1]. Therefore, early clinical decision-making is crucial. Abdominal ultrasound, especially the Focused Assessment with Sonography for Trauma (FAST) and computed tomography, play a very important role in the early assessment of traumatized patients [3]. FAST is well-established and recommended in detecting free intra-abdominal fluid [4]. Further diagnostics are needed in more severe and challenging cases, especially among patients with preexisting intra-abdominal fluid collections, for example, in the presence of ascites due to hepatic cirrhosis. Differentiation between ascites and an acute intra-abdominal hemorrhage, which would possibly need immediate intervention, has to be achieved swiftly. Nowadays, Computed Tomography (CT) has become a widely used and critical imaging technology for the detection of intra-abdominal injuries in patients suffering from severe abdominal trauma. In CT imaging, attenuation values measured by Hounsfield Units (HU) are used to further distinguish between different types of intra-abdominal fluid collections [5]. The aim of this study was to investigate the accuracy of CT imaging in differentiating simple ascites from acute intra-abdominal hemorrhage in patients with preexisting ascites.

## 2. Experimental Section

### 2.1. Study Design

This project was planned and conducted as an experimental study. Ascites was collected from a 62-year-old male patient, suffering from hepatic cirrhosis of Child–Pugh class C, during a clinical routine. In addition, blood was taken from the ascites donor and two further blood group identical voluntary donors. Written consent was given by all three voluntary donors. Ethical approval was obtained by the Ethics Committee of the University of Regensburg (17-783-101).

### 2.2. Sample Processing

Immediately after collecting the ascites and whole blood samples, three serial dilutions were prepared. The three blood samples were gradually diluted with the ascites, so that there were ten dilution steps for each blood sample, creating blood-to-ascites-ratios between 1:1 and 1:1024, before undergoing further processing (Figure 1).

### 2.3. Laboratory-Chemical Analysis

All samples (ascites, three whole blood samples and three serial dilutions with ten steps each) were pipetted into micro reaction vessels (volume 2.0 mL). Laboratory-chemical analysis was performed for all samples, measuring total protein, albumin, leukocytes, erythrocytes, hemoglobin and hematocrit.

### 2.4. Radiological Evaluation

All samples underwent radiological evaluation using CT imaging. Before imaging was obtained, the CT scanner was calibrated according to the manufacturer’s protocol. All studies were performed using a dual source CT scanner (Siemens SOMATOM Definition Flash, Siemens Healthcare GmbH, Erlangen, Germany) with an applied voltage of 120 kV and an effective CT tube current of 350 mAs. Immediately before the first CT examination, the micro reaction vessels were shaken to prevent sedimentation. The samples were positioned within the scanner’s isocenter and scanned in a plastic tray to exclude risk of beam hardening artifacts (Figure 2). Scans were performed at three different points in time: At t_0_ = 0 h or right after, at t_1_ = 1 h after and at t_2_ = 2 h after dilution and pipetting into the micro reaction vessels. The scans were reconstructed with a soft kernel and soft tissue window, and a slice thickness of 1 mm. Density measurements in average HU were performed at two different locations within the micro reaction vessels on the obtained CT data: at R_1_ = 1 cm and R_2_ = 2 cm height from the bottom of the micro reaction vessel, which was measured in the sagittal plane. The measured area itself was defined to be 25 mm^2^ and was determined with the region of interest (ROI) cursor in the axial plane. Furthermore, all measurements (t_0_, t_1_, t_2_ and R_1_, R_2_) were repeated three times. Statistical analysis was performed using mean values of the three measurements. The measurements were performed using the OsiriX MD software package (pixmeo, Bernex, Switzerland).

### 2.5. Statistical Evaluation

All data were analyzed using SPSS version 25 software (SPSS Inc., Chicago, IL, USA). Paired samples t-test and one-way ANOVA in combination with the Bonferroni post hoc test were applied. *p*-values of 0.05 or lower were considered statistically significant.

## 3. Results

### 3.1. Laboratory-Chemical Analysis

Table 1 shows the evaluation of ascites and the three whole blood samples.

The three blood samples showed differences, especially in albumin levels, hemoglobin and hematocrit. Figure 3 displays the mean laboratory-chemical values of the three serial dilutions. A 10-step dilution series was performed on each blood sample diluted with ascites of one donor with blood-to-ascites-ratios from 1:1 to 1:1024.

Hemoglobin measurements resulted in values between 7.5 g/dL and no detectable hemoglobin among 1:512- and 1:1024-ratio-samples, whilst hematocrit was quantified between 21.5 and 0% in the 1:1024-ratio-samples. Albumin levels among the serial dilutions ranged from 21.7 to 7.8 g/L, and total protein concentration from 38.6 to 14.7. Expectedly, by increasing dilution, the composition of the samples gradually equals the pure ascites.

### 3.2. Radiological Evaluation

HU values of ascites, whole blood and all steps in the prepared dilution series were measured using CT. Figure 4 shows the mean HU measurements of the three- dilution series, as described.

A statistically significant difference could only be seen between ascites and whole blood, as well as between ascites and the 1:1-ratio-samples (*p* < 0.001). All further dilutions, ranging from 1:4 to 1:1024, showed no statistically significant difference in average HU values compared to ascites (*p* ≥ 0.42). Whole blood showed significantly higher HU values than ascites and every step of the serial dilutions (*p* < 0.001).

Further statistical analysis of the obtained mean HU measurements by time of measurement (t_0_ = 0 h or right after, at t_1_ = 1 h after and at t_2_ = 2 h, after dilution and pipetting the samples into the micro reaction vessels) was performed. Figure 5 depicts the statistical analysis of measured HU values over time.

Repeated measures ANOVA revealed a rise of HU values over time, which was statistically significant between t_0_ and t_2_ (*p* = 0.01). The mean HU measurement rose from 13.5 at t_0_ to 16.11 at t_2_.

We also evaluated the obtained measurements by localization (R_1_ = 1 cm and R_2_ = 2 cm height from the bottom of the micro reaction vessel, which was set in the sagittal plane) at t_2_, using paired samples *t*-test (Figure 6).

This analysis revealed a statistically significant difference between measured HU values at 1 and 2 cm height (*p* < 0.001). The mean HU measurements from dilution series 1, 2 and 3 at 1 and 2 cm height from the bottom of the micro reaction vessel were 16.52 and 13.78, respectively.

## 4. Discussion

In this study, we tried to simulate the dilution of an acute trauma-associated intra-abdominal hemorrhage in patients with preexisting ascites, performing an in-vitro dilution series. The aim of this project was to evaluate whether CT imaging is suited to differentiate between ascites and concomitant intra-abdominal hemorrhage.

While water has a HU value of 0, blood has been reported with values of 30 HU and above, with clotted blood resulting in higher HU values than fresh blood [6]. Serous fluids, depending on cell purity and protein levels, can show a wide spectrum of HU values. Ascites has been measured with values between 11.0 and 26.4 HU in vitro by Bydder et al. in 1980, closely correlating with protein concentration [5]. Since different types of fluid collections can only be distinguished up to a certain extent using abdominal sonography, decisions regarding further treatment usually rely upon CT imaging results. It is known that the measured attenuation values are influenced by several factors, including individual anatomy, scanner settings (tube voltage and current), photomultiplier gain, possible artefacts and variances in the reconstruction algorithm [5,7].

Low CT attenuation values of acute intra-abdominal hemorrhage have already been described in the past [8]. However, Levine et al. excluded patients with a clinical history or CT graphic evidence of preexisting conditions, such as cirrhosis or abdominal malignancies, that would result in ascites. Patients with suspected hollow organ injuries of bowel, gall or urinary bladder were excluded as well. Because of these exclusion criteria, the effect of dilution was not considered. We believe that our study provides additional information regarding this phenomenon.

The statistical analysis of the obtained mean HU values of ascites, blood and the serial dilution steps revealed that ascites only showed statistically significant lower HU values than whole blood and the 1:1-ratio-sample. All further steps, starting at a ratio of 1:4, did not show a statistically significant difference to pure ascites. These results show that especially small amounts of fresh blood, diluted in larger amounts of preexisting ascites, cannot be reliably detected by analysis of HU in CT imaging. Laboratory-chemical analysis of the different serial dilution steps showed, as expected, a gradual decline especially of albumin-levels, leukocytes, erythrocytes, hemoglobin and hematocrit. These results prove that qualitatively and quantitatively reliable serial dilutions were performed. The impact of these different concentrations on HU measurements has already been described previously [5].

The HU measurements in this project were obtained with the best possible precision. The CT device was calibrated directly before the measurements, all measurements were performed three times in a row and different heights and points in time were measured, to simulate sedimentation. We found a statistically significant rise of HU values over time, and significantly lower values in a different localization of measurement. Furthermore, three different blood samples with different Hb-levels were used.

The fact that, of the three different blood samples, ascites from only one patient was used is one of this project’s limitations. Nevertheless, we tried to avoid unwanted clotting reactions by using, next to whole blood samples of the ascites donor, blood samples with the same blood group as the ascites donor. Moreover, the small size of the micro reaction vessels (2 mL) used in sample processing, and the fact that no surrounding tissue as in a human body was present, have to be mentioned as possible biases. Still, light bleeding would have to be detected as well, and the measurement conditions were identical for all samples. More limitations would be, that in patients suffering from severe trauma, CT is most commonly performed with intravenous contrast medium administration and evaluations are made in conjunction with other intra-abdominal findings such as free peritoneal air or organ laceration. Thus, our results cannot be directly transferred to the setting of acute intraabdominal bleeding. Further clinical studies are required to clarify this question. Despite the limitations listed, we can present a solid model to evaluate the accuracy of CT imaging in detecting minimal amounts of blood, diluted in ascites.

Ultrasound and CT diagnostics are indispensable tools for detecting free intra-abdominal fluid in severely injured patients. As previously mentioned, CT has become the most essential imaging technology to further differentiate this free fluid collection. In the pre-CT era, injuries to solid abdominal organs resulting in intraperitoneal hemorrhage have often been diagnosed using diagnostic peritoneal lavage (DPL), which were introduced by Root et al. in 1965 [9]. Previous studies have shown a sensitivity of DPL between 82.8 and 100% in detecting intra-abdominal injuries [10,11]. In settings, in which sonography and CT are available, DPL is almost no longer used, mainly due to the required expertise and its invasiveness with the risk of further iatrogenic injuries. According to recent polytrauma guidelines, laparotomy is the gold standard in suspected intra-abdominal hemorrhage, especially in critical unstable patients [12]. According to our results, a possible clinical scenario would indicate an emergency laparotomy because of suspected intra-abdominal bleeding due to the CT scan despite the presence of intra-abdominal (non-trauma related) ascites in critical unstable patients as well as vice versa.

## 5. Conclusions

The present results indicate that in the specific collective of patients with preexisting ascites, suffering from blunt abdominal trauma, attenuation values in CT imaging are not distinctive enough to exclude an acute hemorrhage, diluted in ascites, with certainty. Close monitoring with possible early repetition of imaging is necessary in these special cases, to prevent missing intra-abdominal bleeding, which could lead to a crucial delay in therapeutic decision-making. Further clinical studies need to be conducted to prove these in-vitro findings in a clinical setting.

## Figures and Tables

**Figure 1 jcm-10-00076-f001:**
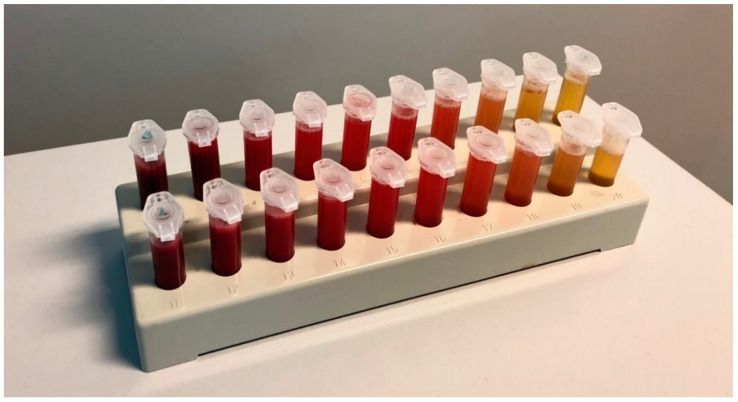
Dilution series. Blood-ascites mixtures ranging from 1:1 (left hand side) to 1:1024 (right hand side).

**Figure 2 jcm-10-00076-f002:**
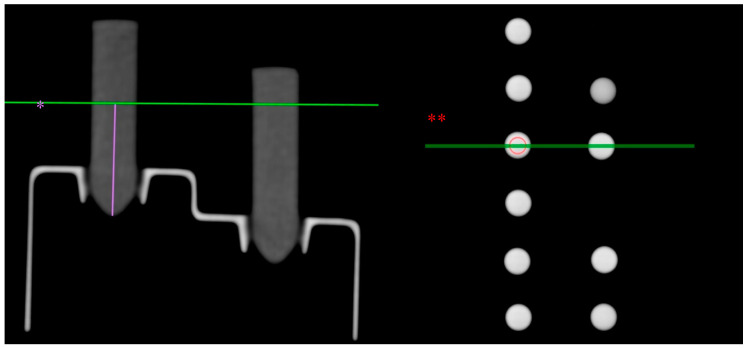
Experimental set-up in Computed Tomography (CT) scans. Measurement of a sample at height R1 in sagittal plane (*); region of interest (ROI) cursor in axial plane (**).

**Figure 3 jcm-10-00076-f003:**
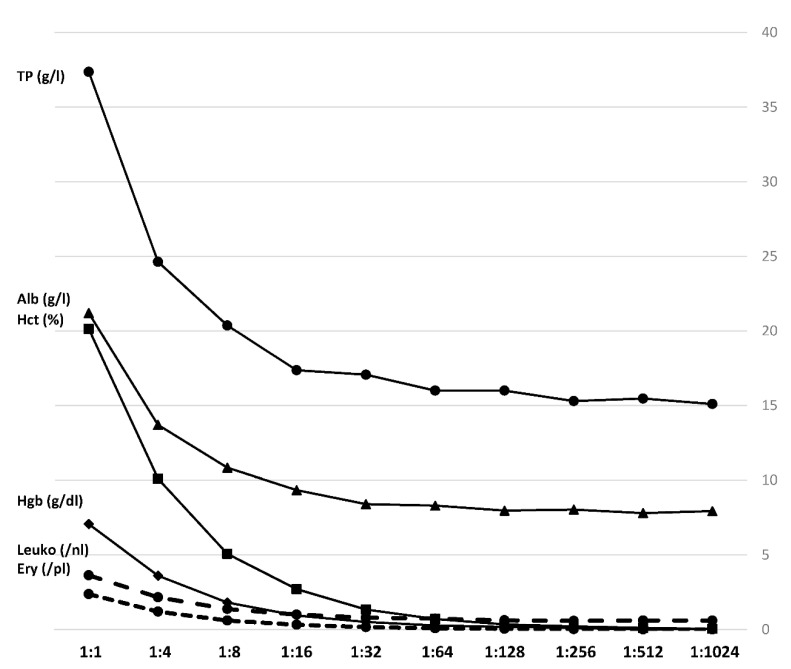
Laboratory-chemical parameters. Mean values of the three samples in a 10-step dilution series. Hgb: Hemoglobin; Hct: hematocrit; TP: total protein; Alb: albumin; Leuko: leukocytes; Ery: erythrocytes.

**Figure 4 jcm-10-00076-f004:**
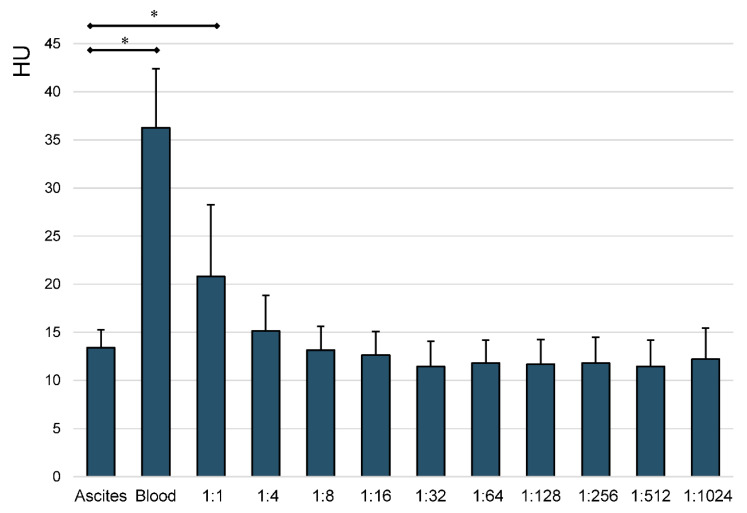
Mean HU measurements of ascites, blood and all steps of the prepared serial dilutions. Whiskers represent standard deviations. Significant differences (* *p* < 0.05) are designated by asterisks.

**Figure 5 jcm-10-00076-f005:**
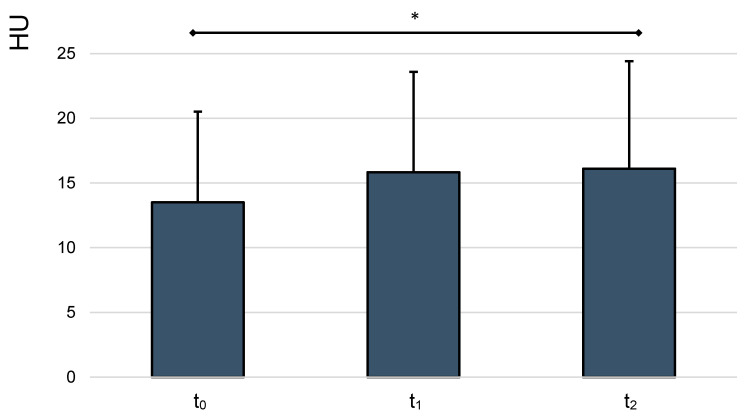
Mean HU values, analyzed over time. Whiskers represent standard deviations. Significant differences (* *p* < 0.05) are designated by asterisks.

**Figure 6 jcm-10-00076-f006:**
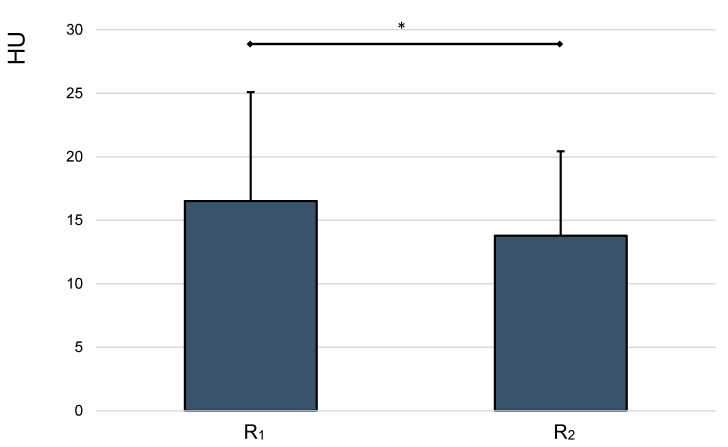
Mean HU values, analyzed by height. Whiskers represent standard deviations. Significant differences (* *p* < 0.05) are designated by asterisks.

**Table 1 jcm-10-00076-t001:** Laboratory-chemical analysis of ascites and blood samples.

	Hgb (g/dL)	Hct (%)	TP (g/L)	Alb (g/L)	Leuko (/nL)	Ery (/pL)
Ascites	0	0	15	7.8	0.5	0
Blood 1	16.2	45.1	84.7	50.4	7.62	5.21
Blood 2	12.8	37	72.5	29.7	7.4	4.45
Blood 3	15.2	42.4	77.7	47.5	4.95	4.77

Hgb: Hemoglobin; Hct: hematocrit; TP: total protein; Alb: albumin; Leuko: leukocytes; Ery: erythrocytes.

## Data Availability

The data presented in this study are available on request from the corresponding author.

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
