# Peer review of "Clinical Value of CT for Differentiation between Ascites and Hemorrhage: An Experimental In-Vitro Study"

_jcm, 2020, doi:10.3390/jcm10010076_

Round 1
Reviewer 1 Report
It is always difficult to comment on the study. The fact that only one patient was examined very much limits this paper's reliability. Obviously the red cell count from bleeding into an abdomen full of ascitic fluid will obviously depend upon the amount of precedant ascites and the amount of haemorrhage. An overly sensitive test will be as unreliable as abdominal puncture, and a less sensitive result will miss significant bleeding. Thus some sort of cut off level would be required. This needs a large number of cases for a stastically significant result.
Author Response
Response to Reviewer 1 Comments:
Point 1: It is always difficult to comment on the study. The fact that only one patient was examined very much limits this paper's reliability. Obviously the red cell count from bleeding into an abdomen full of ascitic fluid will obviously depend upon the amount of precedant ascites and the amount of haemorrhage. An overly sensitive test will be as unreliable as abdominal puncture, and a less sensitive result will miss significant bleeding. Thus some sort of cut off level would be required. This needs a large number of cases for a stastically significant result.
Response 1: Thank you for this comment. We agree, that there are different amounts of ascites and blood admixtures in the clinical situation and our study design exactly mirrors the clinical situation. Therefore, we have tried to reflect the great heterogeneity of the patient population by preparing dilution series. These dilution series appear to be well suited to imitate different patient constellations in terms of blood to ascites ratios and to simulate different amounts of ascites and blood admixtures. We are convinced that despite the simple experimental setup, it could be shown for the first time that intraabdominal hemorrhage in preexisting ascites cannot be safely excluded by a single analysis of Hounsfield Units. In a second step e.g. clinical trial, we hope to obtain more detailed information and, if necessary, can establish cut off levels in a large clinical investigation. Despite the limitations of the study, we hope that the main result, which is examined for the first time, convince you to consider this study for publication.
Reviewer 2 Report
The article holds a firm structure and is interesting in most aspects. In my opinion, there are only minor changes that need to be made.
I am not sure if I recognized a control sample. What does compatible donor mean? I imagine this must be the control sample but the text does not make it clear. Please clarify this.
Taking the clinical significance into account, I would like to read the authors' opinion or consideration of the source of bleeding. The limitations of this otherwise resourceful study contain also the lack of other radiographic signs such as free peritoneal air, solid organ contusions, active bleeding, or other findings a patient would have. Would the results from the tubes change with contrast? What is the role of Multi-modality CT/CE-CT in Trauma?
There are parts that need to be corrected and some proof-reading to be done. e.g. intraabdominal corrected to intra-abdominal and also punctuation.
Furthermore, avoid repeated use of passive voice or present perfect tense and keep sentences simpler. e.g. Overall mortality rates for abdominal trauma victims of up to 24,1% have been reported, could be abdominal trauma victims (are reported to) have up to 24,1% overall mortality rates, etc.
Word order must be taken into account e.g. ...rates especially occur among..., should be ...rates occur especially among...
Author Response
Response to Reviewer 2 Comments:
Changes are highlighted in yellow in the manuscript.
Point 1: The article holds a firm structure and is interesting in most aspects. In my opinion, there are only minor changes that need to be made. I am not sure if I recognized a control sample. What does compatible donor mean? I imagine this must be the control sample but the text does not make it clear. Please clarify this.
Response 1: Thank you for this comment. You are absolutely right, we apologize for the unclear presentation. In addition to one dilution series with patients' blood, we have prepared further dilution series with two blood samples from donors of the same blood group as the ascites donor. Thereby we tried to avoid unwanted clotting reactions between ascites and donor blood to generate valid results. You are absolutely right that the term "blood compatible" can be misleading here, so we have changed the term into "blood group identical donors" at the appropriate places.
Lines 23-24: Ascites was collected during clinical routine. Three serial dilutions, mixing ascites with whole blood samples of the patient and with two blood group identical donors were prepared.
Lines 55-56: In addition, blood was taken from the ascites donor and two further blood group identical voluntary donors.
Lines 180-182: Nevertheless, we tried to avoid unwanted clotting reactions by using, next to whole blood samples of the ascites donor, blood samples with the same blood group as the ascites donor.
Point 2: Taking the clinical significance into account, I would like to read the authors' opinion or consideration of the source of bleeding. The limitations of this otherwise resourceful study contain also the lack of other radiographic signs such as free peritoneal air, solid organ contusions, active bleeding, or other findings a patient would have. Would the results from the tubes change with contrast? What is the role of Multi-modality CT/CE-CT in Trauma?
Response 2: Thank you for this comment. We agree that many different points must be taken into account in the diagnostic algorithm of abdominal trauma. The presence of free peritoneal air or organ laceration may confirm the diagnosis of intraabdominal hemorrhage even without determination of Hounsfiled units within free intraabdominal liquid. In some cases, however, no cause of bleeding can be detected even in circulatory unstable patients, which is why a precise differentiation of fluid accumulation must be made, especially in these cases (Jost et al. American Journal of Surgery 2017). If these patients are also patients with pre-existing fluid accumulation, this can be difficult and misleading. The present results show that a pure examination of the Hounsfield Units does not exclude a bleeding with certainty in this collective. We elaborated this point further in the discussion. Our experimental setup cannot represent the influence of contrast medium on the density values of free intra-abdominal fluid accumulation, but we agree that this might change the Hounsfield Units measured to a small extend. In our opinion, this requires future investigations as well as a further clinical study. We have emphasized this point again.
Lines 185-191: One more limitation would be, that in patients, suffering from severe trauma, CT is most commonly performed with intravenous contrast medium administration and evaluation is made in conjunction with other intra-abdominal findings such as free peritoneal air or organ laceration. Thus, our results cannot be directly transferred to the setting of acute intraabdominal bleeding. Further clinical studies are required to clarify this question. Despite the limitations listed, we can present a solid model to evaluate the accuracy of CT-Imaging in detecting particularly small amounts of blood, diluted in ascites.
Point 3: There are parts that need to be corrected and some proof-reading to be done. e.g. intraabdominal corrected to intra-abdominal and also punctuation. Furthermore, avoid repeated use of passive voice or present perfect tense and keep sentences simpler. e.g. Overall mortality rates for abdominal trauma victims of up to 24,1% have been reported, could be abdominal trauma victims (are reported to) have up to 24,1% overall mortality rates, etc.Word order must be taken into account e.g. ...rates especially occur among..., should be ...rates occur especially among...
Response 3: The manuscript was again proof-read and corrected by a native speaker.
Reviewer 3 Report
Introduction
comment about pan-scans in the trauma patient - recent studies (REACT-2 Sierink JC; and Murphy SP, J Emerg Med 2017), that refute the statement that pan-scans improve survival.
the author's reference is correct, but they should be aware that there is also high quality studies saying pan-scanning do not improve survival and outcomes.
Are there any studies suggesting how many trauma patients present with pre-existing intra-abdominal fluid collections? I just wonder about the relevance.
Methods
I understand this is a small study, but it would have been important to have ascitic fluid from more than one patient to see if there are differences in the components, and whether that made any difference. (noted as a limitation)
Otherwise, I feel the methods are sound. I think the authors did a nice job with the measurements of HU at different heights in the vials, and using the serial dilutions.
Results
no major concerns. Presented figures are easy to understand
Discussion
good discussion about their limitations.
My main concern is applicability. I feel these results are real, and are somewhat interesting, but clinically I don't feel they are that relevant. If a trauma patient presents with free fluid on bed-side ultrasound in the trauma room, surgery should be involved; regardless if they are a known liver patient. If the attending physician feels they are stable enough for a CT, then they will get a CT. I don't really see how this study makes a clinical difference, and I don't see it changing clinical practice.
the authors discuss DPL, but don't mention the used of eFAST ultrasound for abdominal injury. CT is the gold standard, but bedside ultrasound is increasingly used early in the care of a trauma patient. Comparisons of U/S to CT should be included.
Conclusions
Strong wording to say you are "convinced" given ascites from one patient and blood from three. Consider changing the wording. I don't disagree that your results suggest you cannot distinguish an acute abdominal hemorrhage in the patient with known ascities, but given the sample size, I feel this is too strongly worded.
Author Response
Response to Reviewer 3 Comments:
Changes are highlighted in yellow in the manuscript.
Point 1: Comment about pan-scans in the trauma patient - recent studies (REACT-2 Sierink JC; and Murphy SP, J Emerg Med 2017), that refute the statement that pan-scansm improve survival.
the author's reference is correct, but they should be aware that there is also high quality studies saying pan-scanning do not improve survival and outcomes.
Response 1: Thank you for this comment. We agree, that our statement simplifies the complex diagnostic algorithm too much. In the publication you quoted, which focuses on the mortality of children in particular when using a WBCT, it is clear that selective CT diagnostics is not associated with increased mortality compared to WBCT diagnostics. A selective CT diagnostic assessment of the severely injured patient is nevertheless part of the treatment algorithm and can improve survival. We have therefore toned down our statement. Please compare the revised manuscript.
Lines 39-41: Abdominal ultrasound, especially the Focused Assessment with Sonography for Trauma (FAST) and computed tomography, play a very important role in the early assessment of traumatized patients (3).
Point 2: Are there any studies suggesting how many trauma patients present with pre-existing intra-abdominal fluid collections? I just wonder about the relevance.
Response 2: Interesting point. A few papers deal in particular with the therapeutic algorithms for the treatment of a so-called “unexplained intra-peritoneal free fluid without solid organ injury” (Jost et al. American Journal of Surgery 2017; Mahmood et al. World J Surg. 2014) and with cirrhotic patients with blunt abdominal trauma (Lin et al. Injury 2012). Although there are no epidemiological data available here, we think that this problem is a regularly observed issue in the treatment of trauma patients with corresponding pre-existing conditions especially in the aging population as seen in industrial countries..
Point 3: I understand this is a small study, but it would have been important to have ascitic fluid from more than one patient to see if there are differences in the components, and whether that made any difference. (noted as a limitation) Otherwise, I feel the methods are sound. I think the authors did a nice job with the measurements of HU at different heights in the vials, and using the serial dilutions.
Response 3: Thank you very much for the comment, we have tried to simulate different constellations by making a dilution series to take the heterogeneous collective, that we face in clinical routine, into account. Nevertheless, we are pleased that our methodology convinced you despite the weaknesses mentioned in the limitation section.
Point 4: good discussion about their limitations. My main concern is applicability. I feel these results are real, and are somewhat interesting, but clinically I don't feel they are that relevant. If a trauma patient presents with free fluid on bed-side ultrasound in the trauma room, surgery should be involved; regardless if they are a known liver patient. If the attending physician feels they are stable enough for a CT, then they will get a CT. I don't really see how this study makes a clinical difference, and I don't see it changing clinical practice. The authors discuss DPL, but don't mention the used of eFAST ultrasound for abdominal injury. CT is the gold standard, but bedside ultrasound is increasingly used early in the care of a trauma patient. Comparisons of U/S to CT should be included.
Response 4: Thanks for the comment. We believe that the results presented in this study are highly relevant for the diagnosis of critically injured patients. Especially in cases that are hemodynamically unstable without evident organ laceration or source of bleeding intraabdominally, but with intraabdominal fluid formation, the clinical decision making between surgical intervention or any other cause for impaired hemodynamics e.g. heart failure, can be straightened by use of Hounsfield Unit measurement of the intraabdominal free fluid. In our opinion, ultrasound diagnostics is an indispensable tool, especially in the early trauma-care phase, to diagnose free intraperitoneal fluid collections. If free fluid is seen in FAST diagnostics, the question of etiology arises. In the specific patient population this work deals with, there exist a multitude of possible causes for the presence of intra-abdominal fluid after trauma. Ultrasound diagnostics is only of limited use in differentiating between hemorrhage or perhaps a fluid formation that already existed before the trauma (e.g. ascites). Currently, a density value analysis of the fluid accumulation in CT is usually performed for further evaluation. The main finding of the present study is, that even if there is a clinical suspicion of bleeding, an inconspicuous density value cannot rule out the possibility of bleeding and it is imperative to continue the search for bleeding sources. In our discussion, we have emphasized the importance of ultrasound diagnostics, because we believe that this diagnostic method is indispensable for the initial treatment of trauma patients, but is limited in special cases.
Lines 192-194: Ultrasound and CT diagnostics are indispensable tools for detecting free intra-abdominal fluid in severely injured patients. As already mentioned, CT has become probably the most important imaging technology to further differentiate this free fluid collection.
Point 5: Strong wording to say you are "convinced" given ascites from one patient and blood from three. Consider changing the wording. I don't disagree that your results suggest you cannot distinguish an acute abdominal hemorrhage in the patient with known ascities, but given the sample size, I feel this is too strongly worded.
Response 5: Thank you very much for your comment. We have changed the statement to:
Lines 207-209: The present results indicate that in the specific collective of patients with preexisting ascites suffering from blunt abdominal trauma, attenuation values in CT-imaging are not distinctive enough to exclude an acute hemorrhage, diluted in ascites, with certainty.
Round 2
Reviewer 1 Report
Needs some tweaking of the English. Also whether American or UK orthography is preferred.